# Prognostic Nutritional Index after Chemoradiotherapy Was the Strongest Prognostic Predictor among Biological and Conditional Factors in Localized Pancreatic Ductal Adenocarcinoma Patients

**DOI:** 10.3390/cancers11040514

**Published:** 2019-04-10

**Authors:** Ken Ichikawa, Shugo Mizuno, Aoi Hayasaki, Masashi Kishiwada, Takehiro Fujii, Yusuke Iizawa, Hiroyuki Kato, Akihiro Tanemura, Yasuhiro Murata, Yoshinori Azumi, Naohisa Kuriyama, Masanobu Usui, Hiroyuki Sakurai, Shuji Isaji

**Affiliations:** Hepatobiliary Pancreatic and Transplant Surgery, Mie University School of Medicine, 2-174 Edobashi, Tsu, Mie 514-8507, Japan; keichikawa@clin.medic.mie-u.ac.jp (K.I.); a-hayasaki@clin.medic.mie-u.ac.jp (A.H.); kishiwad@clin.medic.mie-u.ac.jp (M.K.); t-fujii@clin.medic.mie-u.ac.jp (T.F.); uskm007@clin.medic.mie-u.ac.jp (Y.I.); khmnh0610@clin.medic.mie-u.ac.jp (H.K.); iorichan@clin.medic.mie-u.ac.jp (A.T.); yasumura@clin.medic.mie-u.ac.jp (Y.M.); azu1121@clin.medic.mie-u.ac.jp (Y.A.); naokun@clin.medic.mie-u.ac.jp (N.K.); m-usui@clin.medic.mie-u.ac.jp (M.U.); hirodon@clin.medic.mie-u.ac.jp (H.S.); shujiisaji1@mac.com (S.I.)

**Keywords:** chemoradiation therapy, pancreatic ductal adenocarcinoma, locally advanced PDAC, PNI, immunonutritional/inflammatory markers

## Abstract

*Background*: In many malignancies, including pancreatic ductal adenocarcinoma (PDAC), host-related inflammatory/immunonutritional markers, such as the prognostic nutritional index (PNI), modified Glasgow prognostic score (mGPS), and C-reactive protein (CRP)/albumin ratio are reported to be prognostic factors. However, the prognostic influence of these factors before and after chemoradiotherapy (CRT) has not been studied in PDAC patients. *Methods:* Of 261 consecutive PDAC patients who were scheduled for CRT with gemcitabine or S1 plus gemcitabine between February 2005 and December 2015, participants in this study were 176 who completed CRT and had full data available on inflammatory/immunonutritional markers as well as on anatomical and biological factors for the investigation of prognostic/predictive factors. *Results*: In multivariate analysis, the significant prognostic factors were RECIST classification, cT category, performance status, post-CRT carcinoembryonic antigen, post-CRT C-reactive protein/albumin ratio, post-CRT mGPS, and post-CRT PNI. Post-CRT PNI (cut-off value, 39) was the strongest host-related prognostic factor according to the *p*-value. In the patients who underwent resection after CRT, median survival time (MST) was significantly shorter in the 12 patients with low PNI (<39) than in the 97 with high PNI (≥39), at 15.5 months versus 27.2 months, respectively (*p* = 0.0016). In the patients who did not undergo resection, MST was only 8.9 months in those with low PNI and 12.3 months in those with high PNI (*p* < 0.0001), and thus was similar to that of the resected patients with low PNI. *Conclusions*: Post-CRT PNI was the strongest prognostic/predictive indicator among the independent biological and conditional prognostic factors in PDAC patients who underwent CRT.

## 1. Introduction

Because pancreatic ductal adenocarcinoma (PDAC) is a systemic disease at the time of diagnosis, PDAC patients with locally advanced disease require multimodal treatments, including chemotherapy and chemoradiation therapy (CRT), in order to improve their long-term survival [1]. However, the prognosis of patients with PDAC remains unfavorable in comparison with many other types of cancer, even after the introduction of multimodal treatments. The 2017 international consensus statement on borderline resectable (BR)-PDAC provided a new definition that was based on three distinct dimensions—anatomical, biological, and conditional—where not only the anatomical dimension but also the biological and conditional dimensions should be taken into account when deciding on treatment strategies [2]. The biological criteria include the presence of possible or latent metastatic disease, and the conditional criteria include suboptimal performance status (PS) and/or the presence of severe medical comorbidities. According to the anatomical definition of BR-PDAC, achieving margin-negative (R0) resection is difficult without any preoperative treatment, such as neoadjuvant chemotherapy and/or radiotherapy. According to the biological definition, there is a possibility of the extra-pancreatic metastatic disease even in a disease that is potentially resectable based on the anatomical criteria. According to the conditional definition, the patient is at high risk for postoperative morbidity or mortality because of host-related factors, including PS and comorbidities, but has a disease that is anatomically potentially resectable [3].

With regard to host-related conditional factors, inflammatory/immunonutritional factors, as well as PS and comorbidities, are associated with intolerance to neoadjuvant therapy, postoperative morbidity/mortality, and prognosis in patients with PDAC [2]. However, assessment of PS is subjective and reflects functional status at a specific point in time. More objective criteria, such as the neutrophil-to-lymphocyte ratio (NLR), platelet-to-lymphocyte ratio (PLR), C-reactive protein/albumin (CRP/Alb) ratio, modified Glasgow prognostic score (mGPS), and Onodera’s prognostic nutritional index (PNI), have been reported to predict the prognosis of patients with various malignancies, including those with PDAC [4,5,6,7,8].

The PNI was originally reported by Buzby et al. [9] in 1980 to predict postoperative complications after abdominal and thoracic surgery. Subsequently, in 1984 Onodera et al. modified it to make it much simpler to use [10]. In addition to its use as a predictor of postoperative complications, Onodera’s PNI has been widely reported since 2010 to be a prognostic/predictive factor in various malignancies, including PDAC [8,11,12,13,14]. Although neoadjuvant treatment is required for patients with advanced PDAC, including BR-PDAC, according to a prospective randomized controlled trial [15], there have been no reports on inflammatory/immunonutritional factors, including PNI, as prognostic indicators in PDAC patients who undergo CRT.

The aim of this study was to evaluate the pre-CRT and post-CRT inflammatory/immunonutritional markers affecting surgical outcome/prognosis and to assess the appropriate cut-off value for these markers in PDAC patients who underwent CRT, with emphasis on the prognostic significance of the PNI.

## 2. Materials and Methods

Between February 2005 and December 2015, 261 consecutive PDAC patients with a cytological or histological diagnosis of localized PDAC determined by 64-slice multidetector computed tomography (MDCT) received our previously reported regimen of chemoradiotherapy followed by surgery (CRTS) [3,16,17,18]. When CRT was completed, we re-evaluated the patient’s condition. Patients who were considered to have a contraindication to curative-intent surgery continued to receive chemotherapy and then underwent a restaging evaluation by MDCT. Ten patients who did not complete CRT, 10 who declined re-evaluation, and two who underwent surgery at another hospital were excluded, leaving 239 (91.6%) of the original 261 patients available for re-evaluation after completion of CRT (Figure 1). A further 10 patients were excluded because they developed other conditions and could not continue subsequent treatment (four developed intractable cholangitis, two gastrointestinal bleeding, two intestinal obstruction, one intractable gastric ulcer, and one respiratory dysfunction. A further 53 patients who had insufficient clinical data available to evaluate immunonutritional/inflammatory markers before and after CRT were also excluded, leaving 176 patients available for analysis in the study.

Computed tomography was performed according to a defined pancreas protocol as four-phasic contrast-enhanced MDCT with thin slices at intervals of 1 mm. Patients were excluded at the time of enrollment if they showed evidence of distant metastasis. All participants provided written informed consent for chemoradiotherapy for PDAC in the study. The study protocol, which used an opt-out approach, was approved by the medical ethics committee of Mie University Hospital (No. H2018-040) and was performed in accordance with the tenets of the 1964 Declaration of Helsinki. Clinical and follow-up information was extracted from a prospectively maintained database at the Department of Hepato-Biliary Pancreatic and Transplant Surgery, Mie University, and verified by reviewing patient medical records. The day of the final follow-up was 30 June 2017.

### 2.1. CRT Protocol and Reassessment

Eighty-one of the 176 PDAC patients enrolled in this study received gemcitabine (GEM)-based chemoradiation therapy (G-CRT) between February 2005 and October 2011, and the remaining 95 received S-1 + G-based CRT (GS-CRT) between November 2011 and December 2015. S-1 is an orally administered agent that contains tegafur, gimeracil, and oteracil [19] and appears to be at least equivalent to or even more active than 5-fluorouracil when combined with radiotherapy for locally advanced PDAC [20,21]. The patients who underwent G-CRT received an infusion of GEM at a dose of 800 mg/m^2^ on days 1, 8, 22, and 29 for one cycle [16,17,22,23] and those who underwent GS-CRT received S-1 orally twice daily at a dose of 60 mg/m^2^/day on days 1–21 of a 28-day cycle and an infusion of GEM at a dose of 600 mg/m^2^ on days 8 and 22 for one cycle. All patients were treated with three-dimensional conformal radiotherapy using the four-field box technique from directions that avoided exposure of the kidney to unnecessary radiation, as it was considered an organ at risk. The gross tumor volume, including the main tumor and lymph nodes with diameter >1 cm, was defined based on the CT images as follows. The clinical target volume was defined as the gross tumor volume plus a 5-mm margin in all directions. Basically, the planning target volume was defined as the clinical target volume plus a 5-mm margin with an additional 10-mm margin added in the cranial-caudal direction. The total radiation dose delivered was 45–50, 4 Gy in 25–28 fractions (five fractions/week). The patients were reassessed at four to six weeks after CRT. When we determined that curative-intent resection was possible, they were scheduled to undergo pancreatectomy. Patients who were unsuitable for curative-intent surgery on reassessment continued receiving chemotherapy (GEM or S-1 + GEM) and underwent restaging evaluation by MDCT at three- to four-month intervals. For this study, patients received no systematic nutritional support before or after the intervention.

### 2.2. Indication for Pancreatectomy and Surgical Procedure

At the time of re-evaluation, especially in the case of locally advanced unresectable disease (UR-LA), we determined that curative-intent resection was possible when there were no MDCT findings of stenosis or change in the shape of the celiac trunk or superior mesenteric artery and no MDCT findings of metastatic lesions in distant organs [16,24]. Pancreaticoduodenectomy (PD) or distal pancreatectomy (DP) was then performed. Resection and reconstruction of the portal vein (PV) and superior mesenteric vein (SMV) were performed when the surgeon could not separate the pancreatic head or the uncinate process from these vessels without leaving gross tumor on the vessel. When limited involvement of the common hepatic artery was identified, a segmental resection of this vessel was performed with primary anastomosis. Patients who had an unresectable disease at surgery, which was usually due to the presence of distant metastasis, underwent surgical bypass, as clinically indicated. Patients who were considered to have a contraindication to curative-intent surgery continued to receive chemotherapy (G or GS) and were reevaluated by MDCT after two cycles of additional chemotherapy.

### 2.3. Postoperative Treatment and Follow-Up

From six weeks after resection, we planned to start the postoperative chemotherapy regimen and continue it for at least six months: from February 2005 to May 2013, GEM was administered at a dose of 800 mg/m^2^ biweekly, and from June 2013 to December 2015, S-1 was administered orally twice daily at a dose of 60 mg/m^2^/day on days 1–14 of a 21-day cycle. Depending on how well the patient tolerated the postoperative chemotherapy regimen, we changed the regimen from GEM to S1 or vice versa. After pancreatectomy, all patients were evaluated by physical examination every month, by laboratory tests every two or three months (including for serum carcinoembryonic antigen [CEA] and carbohydrate antigen 19-9 [CA 19-9] levels), and 4-phasic contrast-enhanced MDCT every three months for a period of two years and at six-month intervals thereafter. If serum tumor marker levels increased, the patients were immediately evaluated by MDCT. Sites of recurrent disease were documented at the time of initial recurrence.

### 2.4. Assessment of Anatomical, Biological, and Conditional Factors Before and After CRT

According to the 2017 international consensus statement [2,3], we assessed pre-CRT and post-CRT anatomical, biological, and conditional factors as well as pre-CRT characteristics (e.g., age, sex, body mass index (BMI), tumor location, and radiological response to CRT) according to the response evaluation criteria in solid tumors (RECIST) version 1.1 classification [25]. Post-CRT factors were evaluated at around four weeks after completion of radiation therapy. Classification of resectability and clinical T category were used for evaluation of anatomical factors. The TNM classification and resectability according to the MDCT findings were defined in accordance with the General Rules for the Study of Pancreatic Cancer (4th edition, in English) established by the Japan Pancreas Society (JPS) [26]. The T category according to the JPS is the same as that in the UICC 7th edition [27]. The N category is classified according to the number of positive lymph node metastases (N0, no regional lymph node metastasis, N1a, metastases in 1–3 regional lymph nodes, and N1b, metastases in ≥4 regional lymph nodes). The resectability of PDAC according to the JPS criteria is classified into three groups: resectable; borderline resectable (subclassified into BR-PV [SMV/PV involvement alone] and BR-A [arterial involvement]) and unresectable.

Clinical N category and CEA and CA 19-9 levels were used to evaluate the biological factors. Regional lymph node metastasis could not be diagnosed by biopsy or PET-CT in the present study, so lymph node metastasis was determined according to the diagnostic CT criteria for lymph node metastasis from the Classification of Pancreatic Cancer by the JPS. On dynamic MDCT, an enlarged node measuring ≥10 mm in the shorter diameter that included a low absorption area suggesting areas of necrosis was diagnosed as metastasis. In patients with obstructive jaundice, drainage was achieved using endoscopic retrograde biliary drainage, endoscopic nasobiliary drainage, or percutaneous transhepatic biliary drainage before CRT. CA 19-9 level may not be an accurate reflection of disease status in patients who express the Le(a-b-) genotype [28]. However, genotyping was not performed routinely, and thus, we could not identify patients with Le(a-b-) genotype. These patients usually presented with values lower than the assay sensitivity threshold (1 U/mL). Eleven of the 176 patients (6.3%) had no detectable serum CA 19-9.

Evaluation of conditional factors included the following: Eastern Cooperative Oncology Group PS [29]; inflammatory/immunonutritional markers including the serum albumin level, lymphocyte count, platelet count, CRP level, CRP/Alb ratio [30], NLR [4], PLR [5], Glasgow prognostic score (GPS) [31], modified GPS (mGPS) [32], and PNI according to the Onodera definition [10]. CRP/Alb ratio, NLR, PLR, GPS, mGPS, and PNI for each patient were determined as follows: CRP/Alb ratio (CRP/albumin in mg/dL/g/dL), NLR (neutrophils/lymphocytes in μL/μL); PLR (platelets/lymphocytes in μL/μL), GPS (score 2 [CRP level > 1.0 mg/dL and serum albumin level < 3.5 g/dL], score 1 [CRP level > 1.0 mg/dL or serum albumin level < 3.5 g/dL], and score 0 [CRP level ≤ 1.0 mg/dL and serum albumin level ≥ 3.5 g/dL]); mGPS (score 2 [CRP level > 1.0 mg/dL and serum albumin level < 3.5 g/dL], score 1 [CRP > 1.0 mg/dL], score 0 [CRP ≤ 1.0 mg/dL]); and PNI (10 × albumin [g/dL] + 0.005 × total lymphocyte count [per mm^3^]).

### 2.5. Statistical Analysis

Continuous and categorical variables were expressed as median (range) and were compared using the Mann–Whitney *U* test and the Chi-square test. In all patients who were reassessed, the date of the initial treatment was chosen as the starting point for measurement of survival. Patients who were alive or had died of a cause other than PDAC were censored for analysis of disease-specific survival (DSS). Survival was calculated using the Kaplan–Meier method and was compared between the groups using the log-rank test. The day of the final follow-up was 31 December 2017, at which time there was no loss of follow-up. Factors affecting DSS were analyzed using the multivariate Cox proportional hazards model. Individual variables with a significance of *p* < 0.05 in the univariate Cox proportional hazards model were selected for inclusion in the multivariate analysis. Variables with a significance of *p* < 0.05 were selected for multivariate analysis. Cut-off values for post-CRT CEA, post-CRT CRP, and post-CRT PNI were determined using a web-based software tool (Cut-off Finder; http://molpath.charite.de/cutoff), and all variables were dichotomized for the analyses. All statistical analysis was performed using IBM SPSS Statistics for Macintosh, (version 24; IBM Corp., Armonk, NY, USA). A *p*-value < 0.05 was considered statistically significant.

## 3. Results

### 3.1. Patient Characteristics

The median age of the 176 PDAC patients was 67 (range 41–86) years, 116 (65.9%) were male and 60 (34.1%) were female. They were classified into the three resectability groups based on the MDCT findings at the initial visit to our hospital: resectable, *n* = 38, borderline resectable, *n* = 44, and locally advanced unresectable, *n* = 94. According to the TN classification, there were 61 patients (34.7%) in cT3 and 115 (65.3%) in cT4, there were 124 (70.5%) in N0, 47 (26.7%) in cN1a, and five (2.8%) in cN1b. Among 176 patients, 109 (61.9%) underwent pancreatectomy and 67 (38.1%) did not.

The pre-CRT factors in the patients who underwent CRT followed by resection or non-resection are shown in Table 1. In terms of the anatomical factors, there was a significantly lower incidence of UR-LA and cT4 in resected patients than in non-resected patients (44.0% vs. 68.7% and 54.1% vs. 83.6%, *p* = 0.002 and *p* < 0.001, respectively). There were no significant differences in the pre-CRT biological or conditional factors between the resected and non-resected patients. Post-CRT factors are shown in Table 2. The total dose of radiation was significantly higher in non-resected patients than in resected patients (*p* = 0.016). Compared with the non-resected patients, the resected patients had a significantly lower incidence of progressive disease (PD; 6.4% vs. 40.3%, *p* < 0.001), a significantly lower serum post-CRT CA 19-9 level (35.5 IU/L vs. 94.6 IU/L, *p* = 0.005), a significantly higher post-CRT albumin level (3.9 g/dL vs. 3.6 g/dL, *p* < 0.001), a significantly lower post-CRT CRP/Alb ratio (0.037 vs. 0.052, *p* = 0.041), significantly lower post-CRT GPS and mGPS scores (*p* = 0.008 and *p* = 0.041, respectively), and a significantly higher post-CRT PNI value (44.9 vs. 41.6, *p* < 0.001). Biological factors, including CEA and CA 19-9 levels, were significantly decreased after CRT (*p* = 0.012 and *p* < 0.001, respectively), while for the conditional factors (except for the CRP, CRP/Alb ratio, and mGPS), there were significant decreases in albumin, lymphocytes, platelets, and PNI after CRT and significantly increases in the NLR, PLR, and GPS scores (Table 3).

### 3.2. Univariate and Multivariate Analyses of Factors Contributing to DSS

Table 4 shows the exact *p*-values for the power of the prognostic factors contributing to DSS in the univariate and multivariable analyses. The independent prognostic factors contributing to DSS were RECIST classification (*p* = 0.0011), type of chemotherapy (G or GS, *p* = 0.0036), cT category (*p* = 0.000004), post-CRT CEA (*p* = 0.00016), PS (0/1 vs. 2/3, *p* = 0.00014), post-CRT CRP/Alb ratio (*p* = 0.00015), post-CRT PNI (*p* = 0.00002), and post-CRT mGPS (*p* = 0.004). Among the biological and conditional factors, post-CRT PNI was identified as the most powerful prognostic factor according to the *p*-value.

### 3.3. Comparisons of DSS Based on Each Significant Prognostic Factor Identified in Multivariate Analysis

According to the Cut-off Finder, the best cut-off values contributing to DSS were 8.0 ng/mL for post-CRT CEA, 39 for post-CRT PNI, and 0.19 for post-CRT CRP/Alb ratio. Figure 2 shows a comparison of DSS for each significant prognostic factor according to its cut-off value. The RECIST classification and T category (cT3/cT4), which are well-known prognostic factors, seemed to be very useful prognostic predictors. Among the other significant prognostic factors, a cut-off value of 39 for post-CRT PNI was considered to be the most useful index for prognostic prediction.

### 3.4. Role of Post-CRT PNI in Determining the Indication for Surgery

According to the RECIST classification, partial response (PR) and stable disease (SD) are generally considered indications for surgery unless local anatomical factors indicating unresectable disease are found. In total, 102 of 142 patients with PR or SD (71.8%) underwent pancreatectomy and 40 (28.2%) did not. However, in the PD group, only seven of the 34 patients (20.6%) underwent pancreatectomy. We compared DSS between the patients with post-CRT PNI ≥ 39 (high PNI) and those with post-CRT PNI < 39 (low PNI) in the PR+SD group and the PD group (Figure 3). In the PR+SD group, the DSS (median survival time [MST]) in patients with a high PNI (*n* = 102) was significantly better than in those with a low PNI (*n* = 40; 24.9 vs. 9.2 months, *p* < 0.001). In PD group, DSS (MST) did not differ between the patients with a high PNI (*n* = 28) and those with a low PNI (*n* = 6; 10.1 vs. 9.0 months, *p* = 0.131).

With regard to the T category, cT3 PDAC patients are generally considered eligible for curative resection, but those with cT4 disease are usually not. Fifty of 61 patients with cT3 disease (82.0%) underwent resection, while 59 of 115 with cT4 disease (51.3%) underwent resection. We compared DSS between the patients with a high PNI and those with a low PNI in the cT3 group and in the cT4 group (Figure 4). DSS (MST) in the patients with a high PNI was significantly better than in those with a low PNI in both the cT3 and cT4 groups (30.4 vs. 14.5 months and 19.9 vs. 9.0 months, *p* = 0.003 and *p* < 0.001, respectively). Patients with low PNI had a poor prognosis regardless of cT stage.

Based on the degree of resectability classification, DSS in patients with a high PNI was significantly better than in those with a low PNI in the group with resectable disease (MST, 36.8 vs. 8.9 months, *p* = 0.00015) and in the group with unresectable disease (20.4 vs. 9.2 months, *p* < 0.0001) (Figure 5). Among patients with BR disease, DSS in those with a high PNI was better than in those with low PNI (22.3 months vs. 8 months, *p* = 0.075); however, the difference was not statistically significant. These findings also indicate that patients with a low PNI have a poor prognosis regardless of resectability and cT stage.

When we focused on the resected patients, who underwent pancreatectomy as a curative intent surgery regardless of resectability classification, and non-resected patients, DSS (MST) in the patients with high PNI was significantly better than in those with low PNI in both resected and non-resected patients (27.2 vs. 15.5 months and 12.3 vs. 8.9 months, *p* = 0.0016 and *p* = 0.000025, respectively, Figure 6). Interestingly, prognosis in the 12 patients with a low PNI who underwent pancreatectomy was almost the same as that in the 67 patients who did not undergo pancreatectomy.

### 3.5. Patients Characteristics between Post-CRT PNI ≥ 39 and PNI < 39

Comparisons of the characteristics of patients with a high PNI (*n* = 145) and those with a low PNI (*n* = 31) are shown in Table 5. The two groups did not differ in age, sex, BMI, tumor location, resectability classification, cT category, PS, GS dose during chemotherapy, dose of radiation administered, RECIST classification, or rate of development of metastases after CRT. Significant differences were found in the type of chemotherapy administered (GS chemotherapy was significantly more common in patients with a high PNI), dose of GEM when administered as the sole chemotherapeutic agent (significantly more common in patients with a high PNI), and resection rate (significantly higher in those with a high PNI).

## 4. Discussion

In this study, we evaluated the utility of inflammatory/immunonutritional markers in predicting prognosis after completion of CRT in PDAC patients and found that post-CRT PNI was the strongest prognostic/predictive indicator among the independent biological and conditional prognostic factors, including post-CRT CEA, PS, post-CRT CRP/Alb ratio, and mGPS. PDAC patients with a low PNI had a significantly poorer prognosis than those with typical indications for curative-intent surgery (i.e., a high PNI even when they had cT3 disease, PR or SD according to the RECIST classification, and resectable PDAC according to the JPS classification). However, there was no significant difference between patients with a low PNI and those with a high PNI in terms of age, sex, BMI, tumor location, resectability classification, cT category, PS, chemotherapy drug dose in GS therapy, dose of radiation, RECIST classification, or rate of development of metastases after CRT.

We assessed prognostic factors before and after CRT in PDAC patients on the basis of the three anatomical, biological, and conditional dimensions stated in the 2017 international consensus statement^2^ and found that cT4 was a significantly poor anatomical prognostic factor and post-CRT CEA was a significantly poor biological factor. Resectability classification was one of the significant anatomical prognostic factors in univariate analysis but was no longer significant in multivariate analysis because almost all patients with cT4 disease have BR-A or UR-LA disease

Among the biological factors, neither pre-CRT CEA nor pre-CRT CA 19-9 was a significant prognostic factor in multivariate analysis. However, post-CRT CEA (but not post-CRT CA 19-9) was a significant prognostic factor. In PDAC patients, it is widely recognized that CA 19-9 is a clinically useful biomarker for determining resectability, a prognostic marker after resection, and a predictive marker for response to chemotherapy [33,34]. In contrast, CEA has low sensitivity and specificity for diagnosing PDAC, although it is often used in combination with other tumor markers, such as CA 19-9 [35,36,37]. In patients who underwent pancreatectomy, a combination of preoperative CA 19-9 and CEA effectively improved the prognostic prediction. About 100 U/mL of CA 19-9 and 10 ng/mL of CEA were revealed as being potentially helpful for prediction of prognosis [38]. Lee et al. analyzed the factors associated with survival to determine the value of pretreatment CA 19-9 and CEA levels in assessing the prognosis of PDAC regardless of stage (including stage III in 38% and Stage IV in 39.6%) and treatment modality, and they found that elevated CEA (>5 ng/mL), but not elevated CA 19-9 (>37 U/mL), was an independent risk factor contributing to overall survival [39]. In the present study, 53.4% of patients had UR-LA PDAC, suggesting that post-CRT CEA > 8 ng/mL but not post-CRT CA 19-9 is a significant predictive factor of prognosis. These results indicate that CEA is one of the useful prognostic/predictive markers in patients with advanced PDAC. However, CA 19-9 is still a valuable diagnostic marker in these patients because of its high sensitivity and specificity.

Conditional factors, such as host-related factors, e.g, patient PS and inflammatory/immunonutritional markers, are associated with intolerance to preoperative therapy, postoperative morbidity, and mortality, and a poor prognosis. In our study, significant prognostic factors were PS, post-CRT CRP/Alb ratio, post-CRT mGPS, and post-CRT PNI. As we have previously reported, PS was a significant prognostic factor in patients with resectable and UR-LA PDAC [3]. Tas et al. [40]. also reported that PS0/1 PDAC patients had a significantly better prognosis than those with PS2/3 in all stages. In our present study, however, the number of patients with PS 2/3 is only 10, so PS might not be useful to decide indication for surgery after chemoradiotherapy, comparing to PNI. The CRP/Alb ratio and mGPS are combination indexes that include CRP and albumin, and both are recognized as inflammatory and nutritional markers. The PNI is a combination index consisting of albumin and lymphocytes that is calculated using a formula, and it has been the focus of attention as an immunonutritional marker. Thus, CRP, albumin, and lymphocytes are key inflammatory/immunonutritional markers.

CRP is a well-known acute-phase protein produced by the liver as part of the systemic inflammatory response. Proinflammatory cytokines, including interleukin-6, are produced by the tumor or surrounding cells and stimulate the production of acute-phase reactant proteins in the liver, such as CRP [41,42] Therefore, CRP is associated with malignancy. Albumin is recognized as not only a marker of host-related nutritional status but also a marker of inflammation in patients with various types of cancer. Elevated levels of inflammatory cytokines from tumors increase the demand for amino acids, and patients with cachexia will develop low serum albumin levels. Furthermore, these cytokines, including tumor necrosis factor, increase the permeability of the microvasculature as well as the transcapillary passage of albumin, resulting in decreased serum albumin levels [43,44]. CRP and albumin levels are reported to be associated with host inflammatory-nutritional state and to be good prognostic indicators of various malignancies, including PDAC [45,46,47,48,49]. The observation that cancer tissue is infiltrated by white blood cells, mainly lymphocytes, leads to the theory of cancer immunosurveillance, where lymphocytes are thought to safeguard against cancer by identifying and destroying malignant cells [50]. Interestingly, reduction of peripheral lymphocyte counts after radiation therapy and pathologically low lymphocyte infiltration around tumor cells in resected specimens have been associated with poor prognosis in various malignancies, including rectal cancer, glioblastoma multiforme, and non-small cell lung cancer [51,52,53]. Fogar et al. [54] reported that low peripheral lymphocyte counts associated with survival in patients with UICC-stage IIB or higher PDAC and indicated a cut-off level of 1200/μL.

Fairclough et al. first proposed the CRP/Alb ratio as a predictor of patient outcome in an acute medical assessment unit [30]. Recently, the CRP/Alb ratio has been reported to be a prognostic predictor in patients with hepatocellular carcinoma [55] and patients with UICC stage III or IV PDAC with a cut-off value of 0.54 [6]. In the present study, we also found that the post-CRT CRP/Alb ratio (but not the pre-CRT CRP/Alb ratio) was a significant prognostic factor with a cut-off value of 0.19. Elevated preoperative mGPS was previously found to be independently associated with shorter overall survival after pancreatectomy for PDAC [7]. mGPS is more strongly affected by the serum CRP level than is the GPS. Our study also found post-CRT mGPS, but not post-CRT GPS, to be a significant prognostic factor.

Post-CRT PNI was also identified in this study as an independent prognostic factor and according to the *p*-value was the most powerful of the inflammatory/immunonutritional markers. The PNI, which consists of albumin and lymphocytes, was originally proposed as a preoperative risk factor for complications in patients undergoing gastrointestinal surgery [10], but it has been widely used as not only a marker of nutritional status but also a prognostic predictor in various malignancies, including PDAC [11,12,13,14]. In 2010, Kanda et al. [8] reported that PNI > 45 was one of the significant prognostic factors contributing to overall survival in 268 patients who underwent resection for PDAC. However, they did not analyze other inflammatory/immunonutritional markers, such as CRP, NLR, PLR, or GPS, or biological markers, such as CEA and CA 19-9. Following the report by Kanda et al., there have been several studies focusing on PNI as a prognostic factor in patients with PDAC. The cut-off values for the PNI as prognostic predictor were reported to be 47.3 in patients with locally advanced or metastatic PDAC and 40 in those with resectable PDAC [56,57]. These cut-off values were larger than our PNI cut-off value of <39, which might reflect the fact that the subjects in previous studies did not receive CRT. Furthermore, we used the Cut-off Finder, which is a comprehensive and straightforward web application tool that is expected to enable faster optimization of new diagnostic biomarkers [58]. There have been very few reports regarding the usefulness of the PNI as a prognostic/predictive indicator in PDAC patients who undergo CRT, although the oncological benefits of neoadjuvant treatment for BR-PDAC have been confirmed in a prospective randomized controlled trial [15] and CRT is now common in PDAC patients. To the best of our knowledge, this is the first study to evaluate the anatomical, biological, and conditional factors before and after CRT that contribute to prognosis. The most important finding of this study was that the post-CRT PNI was the strongest prognostic indicator. CRT generally leads to lymphocytopenia because of bone marrow suppression and hypoalbuminemia associated with anorexia or malnutrition. Our results suggest that patients who maintain a high PNI even after CRT can have a good prognosis, especially if they have locally advanced PDAC.

Significant conditional prognostic factors in our study were the PNI, CRP/Alb ratio, and mGPS; all of these indices include albumin, and the only difference is the lymphocyte count and CRP. Lymphocytes, especially T-cells, are thought to play an important role in anti-tumor immunity, and the peripheral lymphocyte count can be biologically relevant in terms of tumor response. Kitayama et al. [59] assessed peripheral lymphocyte count as a biological indicator of pathological complete response in patients who underwent CRT for advanced rectal cancer. They found that peripheral lymphocyte count (but not serum CRP level) was the strongest indicator of tumor response to CRT. Their findings also suggested that because tumor cells usually have a tumor-associated antigen, lymphocytes, especially T-cells, may play a central role in anti-tumor immunity and that the absolute number of host lymphocytes could be biologically relevant for tumor response to CRT. The reason why the PNI is a more powerful prognostic factor than the CRP/Alb ratio and mGPS may be that circulating lymphocyte counts are associated more positively with the tumor response to CRT than is the CRP.

In clinical practice, the surgeon’s decision regarding resection of PDAC is not based solely on anatomical resectability criteria, but also takes into account the biological behavior of PDAC as well as the conditional status of the host, that is, the ability of the patient to withstand the physiological challenge of surgery. The present study shows that the PNI is the most useful indicator of the conditional status of the host. Curative-intent surgery should be considered contraindicated in patients with a low PNI even if the tumor seems to be operable, as shown in the outcomes for patients with PR or SD according to the RECIST classification, those with cT3 according to the cT category, and those with resectable disease according to the resectability classification (Figure 3, Figure 4 and Figure 5).

Early intervention with nutritional support and rehabilitation is important to improve the PNI before surgery, especially in elderly patients. Preoperative oral immunonutrition using a supplemental liquid diet was reported to reduce both the risk of postoperative infectious complications and the length of hospital stay in PDAC patients who underwent pancreaticoduodenectomy for PD [60]. Immunonutrition is thought to alter the production of cytokines and immune function, thereby limiting undesirable excessive perioperative stimulation of the immune and inflammatory cascade, and is associated with an increase in tumor-infiltrating lymphocytes that affect the cancer prognosis [61,62]. Early intervention with nutritional support, including oral immunonutrition, may improve the prognosis of PDAC patients.

## 5. Conclusions

In conclusion, post-CRT PNI was found to be the strongest prognostic/predictive indicator of all the independent biological and conditional prognostic factors, including post-CRT CEA, PS, post-CRT CRP/Alb ratio, and mGPS, in PDAC patients who received CRT. Patients with a low PNI would benefit from early intervention with immunonutritional support during and after CRT as well as in the perioperative period to improve their PNI, even if the tumor is potentially anatomically resectable.

## Figures and Tables

**Figure 1 cancers-11-00514-f001:**
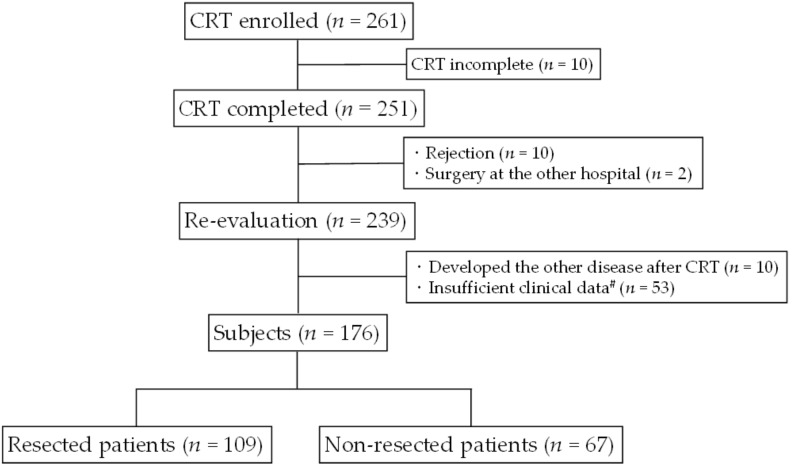
Flow diagram of the PDAC patients in this study. In total, 239 of 261 patients (91.6%) could be re-evaluated after completion of CRT after the exclusion of 10 patients who did not complete CRT, 10 who declined re-evaluation, and two who underwent surgery at another hospital. A further 10 were excluded because they could not continue treatment as they developed other diseases, and another 53 were also excluded because they had insufficient clinical data available before and after CRT. This left 176 patients available for inclusion in the study. ^#^ Lack of data for evaluation of inflammatory/immunonutritional markers, including NLR, PLR, GPS, mGPS, CRP/Alb ratio, and PNI scores before and after chemoradiotherapy. CRP/Alb, C-reactive protein/albumin; CRT, chemoradiotherapy; GPS, Glasgow prognostic score; mGPS, modified Glasgow prognostic score; NLR, neutrophil-to-lymphocyte ratio; PLR, platelet-to-lymphocyte ratio; PDAC, pancreatic ductal adenocarcinoma; mGPS, modified Glasgow prognostic score; PNI, prognostic nutritional index.

**Figure 2 cancers-11-00514-f002:**
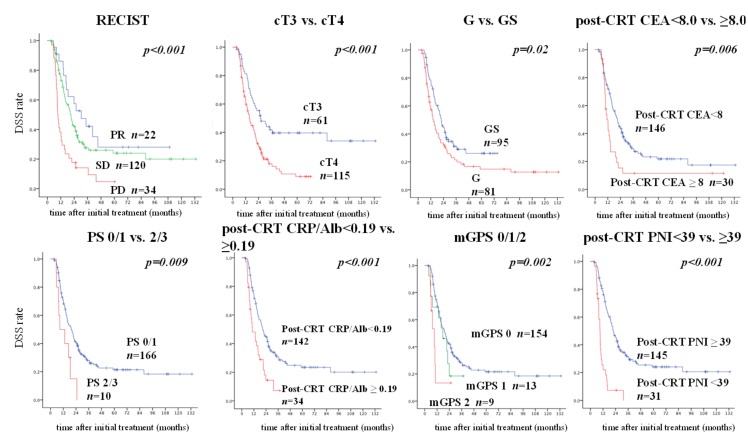
Comparisons of DSS based on each significant prognostic factor in multivariate analysis. The best cut-off value of post-CRT PNI contributing to DSS was 39, that of post-CRT CEA was 8.0, and that of the post-CRT CRP/Alb ratio was 0.19. We compared DSS based on each significant prognostic factor in multivariate analysis using the cut-off value. According to the *p*-value, the post-CRT PNI was the most significant of the immunonutritional/inflammatory markers. CEA, carcinoembryonic antigen; CRP/alb, C-reactive protein/albumin; CRT, chemoradiotherapy; DSS, disease-specific survival; PNI, prognostic nutritional index.

**Figure 3 cancers-11-00514-f003:**
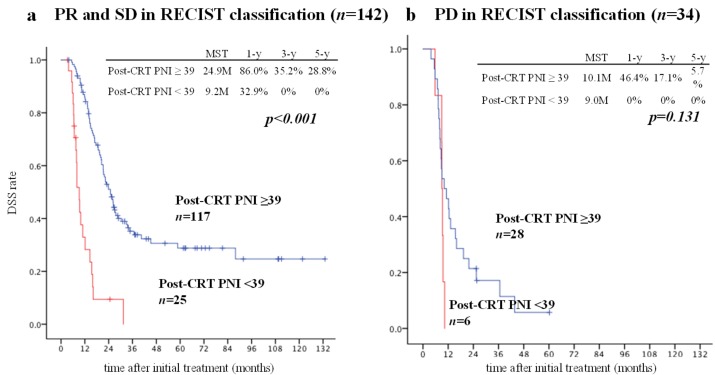
Comparison of DSS between patients with post-CRT PNI ≥ 39 and post-CRT PNI < 39 in patients with PR or SD according to the RECIST classification and in patients with PD. We divided the 176 patients into a PR + SD group (**a**) and a PD group (**b**) according to the RECIST classification. In the PR + SD group, there was a significant difference in DSS between the patients with post-CRT PNI ≥ 39 and < 39 (**a**) (post-CRT PNI ≥ 39 vs. <39: MST, 24.9 vs. 9.2 months, *p* < 0.0001). In the PD group (**b**), the DSS in the patients with post-CRT PNI ≥ 39 was slightly better than that in those with post-CRT PNI < 39. Twelve of 25 patients with a PR or SD and post-CRT PNI < 39 (48.0%) underwent resection. None of the six patients with PD and post-CRT PNI < 39 underwent resection. CRT, chemoradiotherapy; DSS, disease-specific survival; PNI, prognostic nutritional index; PD, progressive disease; PR, partial response; SD, stable disease.

**Figure 4 cancers-11-00514-f004:**
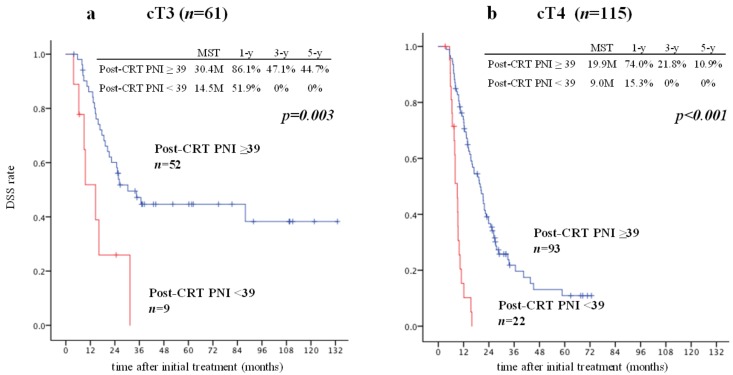
Comparison of DSS between patients with post-CRT PNI ≥ 39 and those with post-CRT PNI < 39 and cT3 or cT4 disease. There was a significant difference in DSS between patients with post-CRT PNI ≥ 39 and those with post-CRT PNI < 39 in the group with cT3 disease (**a**) and cT4 disease (**b**) (post-CRT PNI ≥ 39 vs. <39: MST, 30.4 vs. 14.5 months and 19.9 vs. 9.0 months, *p* = 0.003 and *p* < 0.001, respectively). Five of nine patients with cT3 disease and post-CRT PNI < 39 (55.6%) underwent resection. Seven of 22 patients with cT4 disease and post-CRT PNI < 39 (31.8%) underwent resection. CRT, chemoradiotherapy; DSS, disease-specific survival; PNI, prognostic nutritional index.

**Figure 5 cancers-11-00514-f005:**
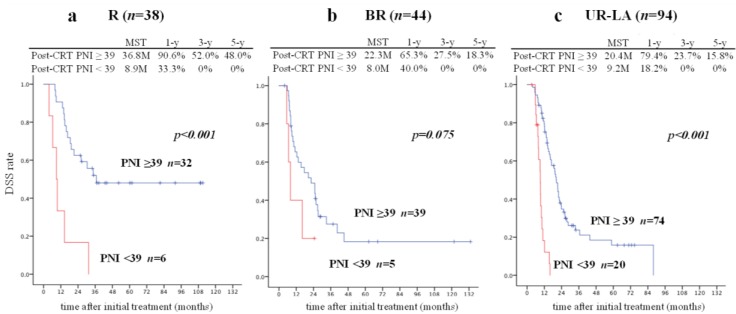
Comparison of DSS in patients who underwent CRT according to whether or not the post-CRT PNI was ≥39 or <39 and resectability classification. When the 176 patients were divided according to PDAC resectability by JPS classification, there was a significant difference in DSS in the resectable (**a**) and UR-LA groups (**c**) (post-CRT PNI ≥ 39 vs. <39: MST, 36.8 vs. 8.9 months and 20.4 vs. 9.2 months, *p* < 0.001 and *p* < 0.001, respectively). DSS in patients with post-CRT PNI ≥ 39 was better than that in patients with post-CRT PNI > 39 in the BR group (**b**) (MST, 22.3 vs. 8 months, *p* = 0.075). Three of six patients with resectable disease and a post-CRT PNI < 39 (50%) underwent resection, as did two of five with BR disease and post-CRT PNI < 39 (40%) and seven of 20 with unresectable disease and post-CRT PNI < 39 (35%). CRT, chemoradiotherapy; DSS, disease-specific survival; JPS, Japan Pancreas Society; MST, median survival time; PDAC, pancreatic ductal adenocarcinoma; PNI, prognostic nutritional index; UR-LA, locally advanced unresectable disease.

**Figure 6 cancers-11-00514-f006:**
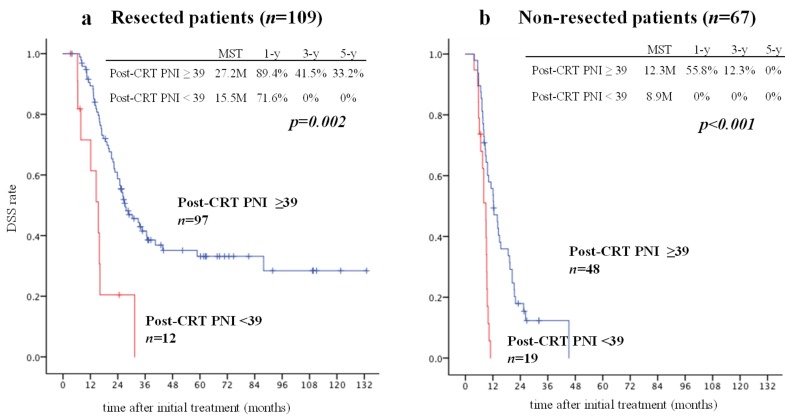
Comparison of DSS between patients with post-CRT PNI ≥ 39 and those post-CRT PNI < 39 who did and did not undergo resection. There was a significant difference in DSS between the patients with post-CRT PNI ≥ 39 and those with post-CRT PNI < 39 who did (**a**) and did not (**b**) undergo resection (post-CRT PNI ≥ 39 vs. <39: MST, 27.2 vs. 15.5 months and 12.3 vs. 8.9 months, *p* = 0.0016 and *p* < 0.001, respectively). CRT, chemoradiotherapy; DSS, disease-specific survival; MST, median survival time; Prognostic Nutritional Index.

**Table 1 cancers-11-00514-t001:** Pre-CRT factors in the patients who underwent CRT followed by resection or non-resection (*n* = 176).

Characteristic	All Patients (*n* = 176)	Resected Patients (*n* = 109)	Non-Resected Patients (*n* = 67)	*p* Value
Age (year)	67 (41–86)	67 (41–86)	68 (43–84)	0.547
Sex (male/female)	116/60	68/41	48/19	0.208
BMI (kg/m^2^)	21.1 (15.1–37.8)	21.1 (15.1–29.4)	21.4(16.4–37.8)	0.792
Tumor location (Ph/Pb/Pt)	123/27/26	82/10/17	41/17/9	0.015
Anatomical factor				
Resectability (R/BR/UR-LA)	38/44/94	32/29/48	6/15/46	0.002
cT category (3/4)	61/115	50/59	11/56	<0.001
Biological factor (Pre-CRT)				
cN category (0/1a/1b)	124/47/5	80/26/3	44/21/2	0.540
CEA (ng/mL)	4.8 (1.0–80.5)	4.4 (1.1–80.5)	5.2 (1.0–54.0)	0.381
CA19-9 (U/mL)	157.6 (0.1–9127)	125.5 (0.1–9127)	186.6 (1.0–7770)	0.221
Conditional factor (Pre-CRT)				
PS 0/1/2/3	106/60/8/2	64/39/5/1	42/21/3/1	0.948
Alb (g/dL)	4.0 (2.9–4.9)	4.0 (2.9–4.7)	4.0 (3.0–4.9)	0.238
Lymphocytes (/μL)	1475 (480–3700)	1510 (540–3700)	1450 (480–3550)	0.562
Platelet (×10^3^/μL)	204 (71–446)	208 (71–446)	198 (96–401)	0.165
CRP (mg/dL)	0.18 (0–9.10)	0.20 (0.01–4.87)	0.16 (0–9.10)	0.869
CRP/Alb	0.051 (0–2.76)	0.052 (0.0025–1.43)	0.036 (0–2.76)	0.792
NLR	2.3 (0.5–9.3)	2.2 (0.9–7.3)	2.6 (0.5–9.3)	0.061
PLR	133.7 (34.4–440)	135.1 (34.43–429.6)	130.1 (38.17–440)	0.717
GPS (0/1/2)	145/25/6	91/15/3	54/10/3	0.750
mGPS (0/1/2)	157/13/6	99/7/3	58/6/3	0.668
PNI	47.2 (34.6–60.8)	47.2 (34.6–58.5)	47.3 (37.3–60.8)	0.625

CRT: chemoradiotherapy, BMI: body mass index, Ph: pancreatic head, Pb: pancreatic body, Pt: pancreatic tail, cT: clinical T, R: resectable, BR: borderline resectable, UR-LA: locally advanced unresectable, cN: clinical N, CEA: carcinoembryonic antigen, CA19-9: carbohydrate antigen 19-9, PS: performance status, CRP: c-reactive protein, NLR: neutrophil-to-lymphocyte ratio, PLR: platelet-to-lymphocyte ratio, GPS: Glasgow Prognostic Score, mGPS: modified Glasgow Prognostic Score, PNI: prognostic nutritional index.

**Table 2 cancers-11-00514-t002:** Post-CRT factors in the patients who underwent CRT followed by resection or non-resection (*n* = 176).

Characteristic	All Patients (*n* = 176)	Resected Patients (*n* = 109)	Non-Resected Patients (*n* = 67)	*p* Value
Chemotherapy (G/GS)	81/95	51/58	30/37	0.795
GEM, mg/m^2^	2688 (496–6425)	2701 (496–6304)	2658 (1166–6425)	0.406
S-1, mg/m^2^	2262 (222–5559)	2304 (222–5232)	2196 (615–5559)	0.366
Radiation 45/50.4 Gy	99/77	69/40	30/37	0.016
RECIST (PR/SD/PD)	22/120/34	19/83/7	3/37/27	<0.001
Biological factor (Post-CRT)				
CEA (ng/mL)	3.9 (1.0–100.6)	3.6 (1.3–31.9)	4.6 (1.0–100.6)	0.131
CA19-9 (U/mL)	47.8 (1.0–13558.8)	35.5 (1.0–1474.9)	94.6 (1.0–13558.8)	0.005
Conditional factor (Post-CRT)				
Alb (g/dL)	3.8 (2.4–4.7)	3.9 (2.4–4.7)	3.6 (2.7–4.7)	<0.001
Lymphocytes (/μL)	995 (380–3210)	1060 (380–3210)	1450 (480–3550)	0.377
Platelet (×10^3^/μL)	181 (41–423)	180 (65–423)	181(41–351)	0.463
CRP (mg/dL)	0.17 (0.01–10.93)	0.14 (0.01–2.76)	0.18 (0.01–10.93)	0.068
CRP/Alb	0.04 (0–2.95)	0.037 (0.0029–0.73)	0.052 (0.003–2.95)	0.041
NLR	3.0 (0.7–14.6)	3.0 (0.9–11.0)	2.9 (0.7–14.6)	0.776
PLR	165.2 (36.0–542.1)	165.1 (47.7–542.1)	171.6 (36.0–503.7)	0.948
GPS (0/1/2)	125/42/9	85/22/2	40/20/7	0.008
mGPS (0/1/2)	154/13/9	96/8/2	55/5/7	0.041
PNI	43.4 (29.0–53.9)	44.9 (29.0–53.5)	41.6 (31.7–53.9)	<0.001

CRT: chemoradiotherapy, G: gemcitabine, GS: gemcitabine and S-1, Gy: gray, RECIST: Response Evaluation Criteria in Solid Tumors, PR: partial response, SD: stable disease, PD: progressive disease, CEA: carcinoembryonic antigen, CA19-9: carbohydrate antigen 19-9, CRP: c-reactive protein, NLR: neutrophil-to-lymphocyte ratio, PLR: platelet-to-lymphocyte ratio, GPS: Glasgow Prognostic Score, mGPS: modified Glasgow Prognostic Score, PNI: prognostic nutritional index.

**Table 3 cancers-11-00514-t003:** Comparison of biological and conditional factors between before and after CRT.

Characteristic	Pre-CRT	Post-CRT	*p* Value
Biological factor			
CEA (ng/mL)	4.8 (1.0–80.5)	3.9 (1.0–100.6)	0.012
CA19-9 (U/mL)	157.6 (0.1–9127)	47.8 (1.0–13558.8)	<0.001
Conditional factor			
Alb (g/dL)	4.0 (2.9–4.9)	3.8 (2.4–4.7)	<0.001
Lymphocytes (/μL)	1475 (480–3700)	995 (380–3210)	<0.001
Platelet (×10^3^/μL)	204 (71–446)	180.5 (41–423)	<0.001
CRP (mg/dL)	0.18 (0.01–9.10)	0.17 (0.01–10.93)	0.771
CRP/Alb	0.051 (0–2.76)	0.04 (0–2.95)	0.967
NLR	2.3 (0.5–9.3)	3.0 (0.7–14.6)	<0.001
PLR	133.7 (34.4–440)	165.2 (35.96–542.1)	0.001
GPS (0/1/2)	145/25/6	125/42/9	0.041
mGPS (0/1/2)	157/13/6	154/13/9	0.594
PNI	47.2 (34.6–60.8)	43.4 (29.0–53.9)	0.001

CEA: carcinoembryonic antigen, CA19-9: carbohydrate antigen 19-9, CRP: c-reactive protein, NLR: neutrophil-to-lymphocyte ratio, PLR: platelet-to-lymphocyte ratio, GPS: Glasgow Prognostic Score, mGPS: modified Glasgow Prognostic Score, PNI: prognostic nutritional index.

**Table 4 cancers-11-00514-t004:** Univariate and multivariate analysis of factors contributing to DSS (*n* = 176).

Characteristic	Univariable, *p*	Multivariable, HR (95% CI)	*p* Value
Age	0.550		
Sex; male vs. female patients	0.476		
BMI	0.117		
Tumor location; Ph vs. Pb vs. Pt	0.245		
RECIST; PR vs. SD vs. PD	0.005		0.0011
Chemotherapy; G vs GS	0.021	1.779 (1.206–2.622)	0.0036
Anatomical factor			
Resectability; R vs. BR vs. UR-LA	0.013		
cT category; 3 vs. 4	0.0002	0.368 (0.241–0.563)	0.000004
Biological factor			
cN category; 0 vs. 1	0.031		
Pre CRT CEA	0.017		
CA19-9	0.095		
Post-CRT CEA	0.001	1.032 (1.015–1.049)	0.00016
CA19-9	0.038		
Conditional factor			
PS; 0/1 vs. 2/3	0.011	0.228 (0.106–0.488)	0.00014
Pre-CRT Alb	0.574		
Lymphocytes	0.231		
Platelet	0.750		
CRP	0.047		
CRP/Alb	0.052		
PNI	0.811		
NLR	0.046		
PLR	0.599		
GPS	0.643		
mGPS	0.530		
Post-CRT Alb	<0.001		
Lymphocytes	0.123		
Platelet	0.701		
CRP	<0.001		
CRP/Alb	<0.001	6.771 (2.515–18.23)	0.00015
PNI	<0.001	0.908 (0.869–0.949)	0.00002
NLR	0.026		
GPS	<0.001		
mGPS	0.004	0.436 (0.246–0.772)	0.004
PLR	0.246		

**Table 5 cancers-11-00514-t005:** Patients characteristics between post-CRT PNI ≥ 39 and PNI < 39.

Characteristic	Post-CRT PNI ≥ 39 (*n* = 145)	Post-CRT PNI < 39 (*n* = 31)	*p* Value
Age	67 (41–86)	70 (49–82)	0.192
Sex (male/female)	91/54	25/6	0.057
BMI	20.8 (15.1–37.8)	22.5 (16.7–27.3)	0.057
Location (Ph/Pb/Pt)	102/22/21	21/5/5	0.865
Resectability (R/BR/UR-LA)	32/39/74	6/5/20	0.343
cT3 vs. cT4	52/93	9/22	0.470
PS 0/1 vs. 2/3	136/9	30/1	0.515
Chemotherapy (G/GS)	61/84	20/11	0.023
G (*n* = 81)			
Dose of GEM (mg/m^2^)	2874 (1890–6425)	2512 (1165–4262)	0.005
GS (*n* = 95)			
Dose of GEM (mg/m^2^)	2610 (496–6304)	2424 (1749–3006)	0.125
Dose of TS-1 (mg/m^2^)	2292 (222–5559)	1991 (1068–3224)	0.106
Dose of Radiation 45/50.4 Gy	81/64	18/13	0.822
RECIST (PR/SD/PD)	21/96/28	1/24/6	0.217
Development of mets after CRT	17 (11.7%) (R:2, BR:8, UR-LA:7)	5 (16.1%) (R:2, UR-LA:3)	0.501
Resected/non-resected (resection rate)	97/48 (66.9%)	12/19 (38.7%)	0.003

CRT: chemoradiotherapy, PNI: prognostic nutritional index, BMI: body mass index, Ph: pancreatic head, Pb: pancreatic body, Pt: pancreatic tail, R: resectable, BR: borderline resectable, UR-LA: locally advanced unresectable, PS: performance status, G: gemcitabine, GS: gemcitabine and S-1, Gy: gray, RECIST: Response Evaluation Criteria in Solid Tumors, PR: partial response, SD: stable disease, PD: progressive disease.

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
