# Peer review of "Prognostic Nutritional Index after Chemoradiotherapy Was the Strongest Prognostic Predictor among Biological and Conditional Factors in Localized Pancreatic Ductal Adenocarcinoma Patients"

_cancers, 2019, doi:10.3390/cancers11040514_

Round 1

Reviewer 1 Report

The Ichikawa study investigated a large cohort of locally advanced pancreatic cancer and radiochemotherapy. The manuscript is written comprehensibly, the illustrations are clear.

Comments:
The P-value should be given uniformly in terms of decimal places.
The investigation of CEA in this setting is not entirely conclusive. The marginal difference of less than 1ng/ml, which is significant, should not be a yardstick for the linear routine and is confusing from my point of view.
It should be clearly described how many patients received nutritional support before and after the intervention.
Is there a correlation between e.g. PNI and ECOG, which one would assume clinically?

Author Response

Reviewer 1

Comments and Suggestions for Authors

The Ichikawa study investigated a large cohort of locally advanced pancreatic cancer and radiochemotherapy. The manuscript is written comprehensibly, the illustrations are clear.

Reply to the comments:

Thank you for your comment.  We appreciated it.

Comments:

The P-value should be given uniformly in terms of decimal places.

Reply to the comments:

We appreciate the reviewer for this helpful comment. 

As you mentioned, we modified uniformly the number of digits of P-value in the content of all figures. However, only in Table 4, we remained the results of multivariate analysis as detailed numerical p-values in order to emphasize the p-value of PNI.

Comments:

The investigation of CEA in this setting is not entirely conclusive. The marginal difference of less than 1ng/ml, which is significant, should not be a yardstick for the linear routine and is confusing from my point of view.

Reply to the comments:

Thank you for your important comment. 

As you mentioned, the marginal difference of less than 1ng/ml of CEA and we might have given a vague impression in Fig 2 because we described CEA <8 vs 8.

The correct cut-off value of post-CRT CEA is 8.0 ng/ml as described in page 8 line 260-261.

We modified from “CEA <8 vs 8” to “CEA <8.0 vs 8.0” in figure 2.

Comments:

It should be clearly described how many patients received nutritional support before and after the intervention.

Reply to the comments:

Thank you for your meaningful comment. 

Actually, during this study period (2005.2 - 2015.12), we had not introduced systematic nutritional support system for PDAC patients. After this study, we try to introduce it for all PDAC patients who plan to undergo chomoraditherapy.

We added the following sentences in page 4 line 137-138:

For this study, patients received no systematic nutritional support before or after the intervention.

Comments:

Is there a correlation between e.g. PNI and ECOG, which one would assume clinically?

Reply to the comments:

Thank your for helpful comments.

As you mentioned, both of PNI and ECOG are important and they are independent factors influencing on DSS. 

However, the number of patients with PS 2/3 are very low (only 10 patients out of 175 (5.7%)) comparing to the number of patients with low PMI (n=31).

Additionally, there are no difference in the incidence of patients with PS 2/3 between high PNI (n = 145) and low PNI (n = 31) (high PNI vs. low PNI: 9/145 (6.2%) vs. 1/30 3/3%), p=0.515) as described in Table 5. These result suggests that PNI is more useful to decide indication for surgery after chemoradiotherapy.

We added the following sentences in page 14 line 396-397: 

In our present study, however, the number of patients with PS 2/3 are only 10, so PS might not be useful to decide indication for surgery after chemoradiotherapy, comparing to PNI.

Additional Comments:

Please provide the blank copy of informed consent from the patients. 

Important, Thank you!

Reply to the comments:

Thank you for your important comment. First of all, we have to apologize that we could not explain exactly regarding the “informed consent from the patients”.

Correctly, it is as follows: They all gave their written informed consent for chemoradiotherapy for PDAC in the study. The study protocol was carried out using an opt-out approach and approved by the medical ethics committee of Mie University Hospital (No.H2018-040),

We have changed the following text from They all gave their written informed consent for inclusion in the study. The study protocol was approved by the medical ethics committee of Mie University Hospital (No.H2018-040),and the study was performed in accordance with the ethical standards established in the 1964 Declaration of Helsinkito All participants provided written informed consent for chemoradiotherapy for PDAC in the study. The study protocol, which used an opt-out approach, was approved by the medical ethics committee of Mie University Hospital (No. H2018-040) and was performed in accordance with the tenets of the 1964 Declaration of Helsinki.,in page 3 line 98-100.

Other corrections:

Our manuscripts were corrected by native English speaker and we putted the certification.

We missed upload table 5 and added it.

Reviewer 2 Report

The authors present a prospective study looking at PNI as a prognostic marker for pancreatic cancer patients treated with chemoradiation. Overall, I think this is an interesting and comprehensive study measuring various prognostic parameters that may lead to treatment decision making. I only have minor comments: 

1. The manuscript is well written overall, but many small grammatical errors are present throughout. 

2. The survival curves include both resected and unresected patients, is it possible that they are separating based on resectablity? especially because resectability was a significant factor in the univariate analysis in table 4. Please provide a clarification- this could be clarified by providing separate survival plots for resectable and unresectable patients. 

Author Response

reviewer 2

Comments and Suggestions for Authors

The authors present a prospective study looking at PNI as a prognostic marker for pancreatic cancer patients treated with chemoradiation. Overall, I think this is an interesting and comprehensive study measuring various prognostic parameters that may lead to treatment decision making. 

Reply to the comments:

Thank you for your comment.  We appreciated it. 

I only have minor comments: 

1. The manuscript is well written overall, but many small grammatical errors are present throughout. 

Reply to the comments:

We appreciate for this helpful comment. 

Our manuscripts were corrected by native English speaker and we putted the certification. 

2. The survival curves include both resected and unresected patients, is it possible that they are separating based on resectablity? especially because resectability was a significant factor in the univariate analysis in table 4. Please provide a clarification- this could be clarified by providing separate survival plots for resectable and unresectable patients. 

Reply to the comments:

Thank you for your comments:

We might confuse you regarding the words “resectable”, “resected”, and “resectablity”.

As to the “resectable” and “resectablity”, we described the following sentences in page 5 line 177-179:

The resectability of PDAC according to the JPS criteria is classified into three groups: resectable; borderline resectable (subclassified into BR-PV [SMV/PV involvement alone] and BR-A [arterial involvement]); and unresectable.

In Figure 5, we revealed comparison of DSS in patients who underwent CRT between patients with post-CRT PNI ≥ 39 and those with post-CRT PNI < 39 according to “resectability classification”. 

In Figure 6, we described comparison of DSS between patients with post-CRT PNI ≥ 39 and those with Post-CRT PNI < 39 in resected and non-resected patients respectively. 

The resected patients means the patients who underwent pancreatectomy as a curative intent surgery regardless of “resectability classification”.

We revised the following sentences in page 11 line 332-336:

When we focused on the resected patients, who underwent pancreatectomy as a curative intent surgery regardless of resectability classification, and non-resected patients, DSS (MST) in the patients with high PNI was significantly better than in those with low PNI in both resected and non-resected patients (27.2 vs. 15.5 months and 12.3 vs. 8.9 months, P = 0.0016 and P = 0.000025, respectively; Fig 6).

Additional Comments:

Please provide the blank copy of informed consent from the patients. 

Important, Thank you!

Reply to the comments:

Thank you for your important comment. First of all, we have to apologize that we could not explain exactly regarding the “informed consent from the patients”.

Correctly, it is as follows: They all gave their written informed consent for chemoradiotherapy for PDAC in the study. The study protocol was carried out using an opt-out approach and approved by the medical ethics committee of Mie University Hospital (No.H2018-040),

We have changed the following text from They all gave their written informed consent for inclusion in the study. The study protocol was approved by the medical ethics committee of Mie University Hospital (No.H2018-040),and the study was performed in accordance with the ethical standards established in the 1964 Declaration of Helsinkito All participants provided written informed consent for chemoradiotherapy for PDAC in the study. The study protocol, which used an opt-out approach, was approved by the medical ethics committee of Mie University Hospital (No. H2018-040) and was performed in accordance with the tenets of the 1964 Declaration of Helsinki.,in page 3 line 98-100.

Other corrections:

We missed upload table 5 and added it.